# Preparation of PVA Fluorescent Gel and Luminescence of Europium Sensitized by Terbium (III)

**DOI:** 10.3390/polym12040893

**Published:** 2020-04-12

**Authors:** Yifan Wei, Zhengquan Fu, Hao Zhao, Ruiqi Liang, Chengyu Wang, Di Wang, Jian Li

**Affiliations:** 1Collage of Material science & engineering, Northeast Forestry University, Harbin 150040, China; ivan98cn@163.com (Y.W.); fzq1999fzq@163.com (Z.F.); chill_blue@163.com (H.Z.); cosmiclrq@163.com (R.L.);; 2Key Laboratory of Biobased Material Science & Technology (Ministry of Education), Northeast Forestry University, Harbin 150040, China

**Keywords:** PVA gel, europium, terbium, sensitization, quenching

## Abstract

Polyvinyl alcohol (PVA) gel has a very wide range of applications in agriculture, military, industry, and other fields. As a widely used water-soluble polymer, PVA has good mechanical properties, excellent spinnability, good hydrophilicity, remarkable physical and chemical stability, good film formation, is non-polluting, and exhibits good natural degradation and biocompatibility. It is an ideal gel preparation material. Incorporation of rare-earth elements into PVA polymers can be used to prepare rare-earth luminescent gel materials. Results show that the luminescent efficiency of complexes is mainly related to their structure, ligand substituents, synergists, and the electronic configuration of doped rare-earth ions. Fluorescent gel films were prepared by adding europium, terbium, and europium/terbium co-doped into PVA, and their fluorescence properties were compared and analyzed. It was found that, in addition to the above factors, the sensitization of terbium to europium, and the fluorescence-quenching effect of hydroxyl groups, will influence the fluorescence properties. This has opened a new route for the application of rare-earth materials and may have value in the field of new materials.

## 1. Introduction

In recent years, there has been a great interest in the development of new synthetic hydrogels and hydrogel composites, which can be attributed to their unique combination of properties, including biocompatibility, permeability, hydrophilicity, and low friction coefficient [1,2,3,4,5]. Polyvinyl alcohol (PVA) also has good light transparency. According to existing studies, PVA can be used as the raw material for the synthesis of artificial corneas [6,7,8]. Most of PVA fluorescent films are used as luminescent materials and detection materials in recent research. One type of luminescent material is embedded with phosphors containing rare earth minerals using soluble sodium silicates (water glass), hydroxyethyl cellulose (HEC) and PVA as adhesives. It was demonstrated that the converter films can be recycled by dissolving the films in water at room temperature for HEC and PVA and at 60 °C for the sodium silicates [9]. By contrast, the preparation of PVA gels, which require only natural cooling, is simple. Fluorescence efficiency was improved by introducing organic ligands. The hydrogels have a unique response time to different ammonia environments through a fluorescence quenching pattern. This process can be identified by the naked eye, so there are potential applications for fluorescent hydrogels in ammonia detection.

At present, there are many surface-treatment methods of PVA, such as silanization, electrospinning, electrochemical oxidation, solution casting, immersion precipitation, and the sol-gel method. Different treatment methods have different effects on the PVA surface [10,11,12,13,14]. The silane coupling agent can not only react with the hydroxyl group in an inorganic substance, but also interact with the long molecule in an organic substance, so it can bridge the two phases and connect them [15,16]. Inorganic material is simple to prepare, and, e.g., wood is modified with rare-earth complexes to produce PVA fluorescent film material, which not only maximizes the characteristics of PVA, but also endows it with new fluorescence properties, so that the modified PVA can be better applied in interior decoration, handicraft production, and other uses.

Rare earth metals are oxytropic, and have a variety of coordination modes. The chelation-metal center binding mode is usually used to regulate the structural changes and exhibit catalytic activity for different epoxy substrates [17]. The effect of ligand-sensitized central ion luminescence is called the antenna effect. The complexes centered on rare earth ions are called luminescence centers, and their increasing concentration will lead to an increase in fluorescence intensity in the absence of other influences. When the complex has a larger conjugate plane and a more-rigid structure, energy loss during ligand transfer is reduced and the fluorescence efficiency increases [18].

The fluorescence properties of rare-earth (lanthanide) elements have become a hot research field. Emission color is largely determined by the lanthanide ion, but also depends on the ligand. Most research in this area has been limited to inorganic compounds [19]. Rare-earth elements can be used as excellent luminescent materials, but their molar absorption coefficient is small, and their luminous efficiency is low. Other scientific experiments have shown that the tetratetralorthosilprivate can be used as the matrix to prepare the gel. The process of preparing the gel is extremely complicated, require many drying steps, and takes a long time [20]. By contrast, the preparation of PVA gels, which require only natural cooling is simple. The effects of temperature, time, Eu/Tb ratio, and complex concentration on fluorescence performance were investigated. The reason for choosing Eu and Tb is that europium and terbium are already widely used as fluorescent materials, and because red and green fluorescence are primary colors, a potentially useful factor for fluorescent materials. In addition, it was also found that terbium had a sensitization effect on europium fluorescence and excessive hydroxyl content had a quenching effect on fluorescence. The Eu/Tb ratio determines when sensitization reaches its maximum.

Other studies on rare earth fluorescence materials, organic ligands were used and PVA was used as the matrix. The fluorescence sensitization of terbium to europium and the fluorescence quenching effect of hydroxyl groups were investigated. In this paper we also study hydrolysis-modified PVA and explore the new technology of PVA inorganic functionalization. Our objective is to provide a basic theoretical foundation that has not been currently established in PVA–rare-earth-complex functional composite luminescence materials research. According to the experimental results, the thermal stability of the modified complex was improved after crosslinking with PVA, so increasing temperature did not affect the fluorescence intensity of the film. Therefore, as a recyclable fluorescent material, this product has potential utilization value.

## 2. Experimental Details

### 2.1. Preparation of Silane-Modified Eu/Tb Co-Doped Complexes

First, aminopropyltriethoxysilane (Huawei chemical co. LTD, Beijing, China) (KH550) (3.5419 g) and phthaloyl chloride (TCI development co. LTD, Shanghai, China) (1.6241 g) were dissolved in dichloromethane (10 mL) separately in beakers (50 mL) as solution 1 and 2. Next, the two solutions were slowly mixed in an ice bath into beaker 1 and sealed, followed by magnetic stirring at room temperature for 2 h. The stirred liquid was placed in an oven at a temperature of 45 °C for 75 min to vaporize the HCl that was produced during the reaction to make sure the complex is neutral. The yellowish viscous liquid was obtained as the silane-modified ligand and dissolved in DMF (10 mL). Then, europium nitrate hexahydrate (Huawei chemical co. LTD, Beijing, China) (0.4461 g) and terbium nitrate hexahydrate (Huawei chemical co. LTD, Beijing, China) (0.4530 g) were dissolved in DMF (10 mL). The two solutions were mixed and stirred in a water bath (70 °C) for 4 h, and silane-modified Eu/Tb co-doped complexes were obtained [21]. Silane-modified europium and terbium complexes were produced in the same way, and, by changing the molar ratio of Eu/Tb, silane-modified Eu/Tb co-doped complexes were produced in different molar ratios (Eu:Tb=1:1, 1:2, 1:3, 1:4, 1:5, 1:6, 1:7, 1:8, 1:9, 2:1, 3:1, 4:1, 5:1, 6:1, 7:1, 8:1, and 9:1). The structural formulas are shown in Figure 1.

### 2.2. Preparation of PVA Fluorescent Gel

First, PVA (Guangfu fine chemical research institute, Tianjin, China) (0.5 g) was dissolved in distilled water (4.5 mL) in a three-neck round-bottom flask (100 mL) and stirred in a water bath (90 °C) to dissolve PVA completely. Next, boric acid (0.05 g) was dissolved in distilled water (1 mL) in a beaker and added into the flask. Boric acid (BA) induced the formation of crosslinking through hydrogen bonding with polymers followed by strong ionic bonding between hydroxyl groups of PVA and borate ions of BA and the solution is still neutral after testing by pH indicator paper [22]. Then, the silane-modified Eu/Tb co-doped complexes (0.04 mol/L) were added in a series of volumes (1, 2, …,9 mL) into the flask. Afterwards, the system reached homogeneity, and was reacted for a series of times (15, 30, 45, 60, 75, 105, and 120 min) at a series of temperatures (45, 50, 55, 60, 65, 70, 75, 80, 85, 90, and 95 °C). After a period of reaction, the solution in the flask was poured into a home-made rectangular glass dish (10 cm × 20 cm) forming a gel and vaporizing the solvent.

### 2.3. Structural Characterization and Performance Testing

The size and morphology of the PVA fluorescent gel were obtained by scanning electron microscopy (SEM) (FEI, Netherlands) using a Quanta instrument operated at an acceleration voltage of 160 kV. The chemical groups of the PVA fluorescent gel were detected by a Fourier-transform infrared (FTIR) spectroscopy (FTIR-650) (Gangdong technology co. LTD, Tianjin, China). Thermogravimetric analysis (TGA) (Netzsch instruments co. LTD, Germany) of samples was conducted using a thermogravimetric analyzer (TG209F3). The X-ray-diffraction (XRD) (FEI, Netherlands) patterns of the gel were analyzed using an XRD-6100 instrument (Shimadzu Corp., Japan). Excitation and emission spectra were recorded using an LS55 fluorescence spectrophotometer (PE, America) at room temperature. The surface chemical composition of PVA fluorescent gel was determined by an energy-dispersive X-ray detector (FEI, Netherlands).

## 3. Results and Discussion

### 3.1. Structural Analysis

The Fourier-transform infrared (FTIR) spectra of the modified Eu^3+^/Tb^3+^ complex, PVA fluorescent gel, and PVA gel presented in Figure 2 contain several peaks, which represent several stretching vibrations—C=O, N–H, and C–Si—at approximately 1715, 1471, and 1070 cm^−1^, respectively. The peak positions and their assignments are presented in Table 1. The bands observed at 1715 and 1741 cm^−1^ are due to the stretching relaxation of the C=O and N–H bonds, respectively.The phthaloyl chloride standard infrared spectrogram, in which the peaks of C=O and C–Cl bonds are supposed to exist at 1770 and 790 cm^−1^, respectively, indicates that the phthaloyl chloride reacted with the silane coupling agent containing amino to form acylamino, proving that the modified ligand exists. The C–Si band observed at approximately 1070 cm^−1^ is attributed to the fact that the C–Si bond did not break during the ligand formation. The band observed at 715 cm^−1^ is the infrared characteristic peak of RE-O (RE=Eu^3+^, Tb^3+^) which reveals that the oxygen in acyl forms a coordinate bond with the rare earth ion. Through infrared-absorption spectral analysis, due to the rare-earth-ion structures being similar in certain kinds of modified complex ligands, a similar infrared characteristic peak is shown.

Figure 3 shows the XRD curve for PVA gel and Eu, Tb, and Tb/Eu fluorescent gels. As shown in Figure 3 one peak around 2θ = 20° appeared, corresponding to the (101) plane of PVA [23]. However, the peak broadened and shifted for the samples containing a complex, and it weakened with the addition of more complexes. Peaks at approximately 2θ = 31° and 45° correspond to the (200) and (220) planes of NaCl, respectively [24].

The morphology of the PVA hydrogel and PVA fluorescent gel was studied by SEM. The SEM images presented in Figure 4 reveal the PVA hydrogel is composed of many tiny particles on the surface. Based on the results of energy-dispersive X-ray-spectroscopy analysis (Table 1), it is suspected that the particles on the surface are NaCl. Table 1 also shows the compositions of the PVA hydrogel and PVA fluorescent gel, and the main components in PVA hydrogel are C and O. The table further shows the detailed contents and reveals that the main components are C and O, which are from the PVA hydrogel. The Cl and Na are from phthaloyl chloride and the silane coupling agent, and they are also attached to the PVA spatial structure. Furthermore, according to Table 1, the mass ratio of europium and terbium to silicon is 3.2, but this is different from the theoretical value, which indicates that the distribution of modified europium/terbium co-doped complexes in PVA films was not uniform.

### 3.2. Fluorescence Spectra

Hydrogels are a kind of material composed of water and a high-molecular polymer possessing a three-dimensional cross-linked network structure. They exhibit good water absorption and water retention and are considered efficient luminescent hosts for rare-earth ions to produce phosphors emitting a variety of colors. The electron transition forms of rare earth ions are f–f, f–d and charge transition. Most rare earths have f–f transitions, which are forbidden transitions. In order to overcome the shortcomings of low molar absorption coefficient and low luminous efficiency, rare earth ions can coordinate with organic ligands with high absorption coefficient to form complexes. The center ion of complex could fluoresce, and the organic ligand transfers the absorbed energy to the center rare earth ion in the form of a non-radiative transition to sensitize it to emit characteristic fluorescence. Figure 5 shows the photoluminescence (PL) excitation spectra of the modified rare-earth-complex (Eu^3+^:Tb^3+^ = 1:1) powders. Figure 5a shows the emission spectra of the modified rare-earth complex upon setting the excitation wavelength to 350 nm. The emission spectrum is dominated by the 5D^4^-7F^6^ (492 nm) and 5D^4^-7F^5^ (547 nm) electron transitions of Tb^3+^ with a maximum at 547 nm. Thus, this peak is selected as the characteristic peak. Figure 5b shows the emission spectra of the modified rare-earth complex upon setting the excitation wavelength to 390 nm. The emission spectrum is dominated by the 5D^0^-7F^1^ (594 nm) and 5D^0^-7F^2^ (619 nm) electron transitions of Eu^3+^ with a maximum at 619 nm [25,26,27,28,29,30,31]. Therefore, 619 nm was selected as the characteristic peak.

Figure 6 presents the fluorescence emission intensities of the PVA fluorescent gel with different concentrations by controlling the temperature (90 °C), reaction time (45 min), and Eu^3+^:Tb^3+^ ratio. The figure indicates that the intensity of the characteristic peak of the PVA fluorescent gel first increases and then decreases with increasing volume of the modified rare-earth complex. The intensity of Eu^3+^ has a maximum at 8 mL and that of Tb^3+^ has a maximum at 7 mL. It is clear that the fluorescence emission intensity of PVA fluorescence gel does not increase linearly with reaction concentration, indicating that fluorescence quenching may occur in the reaction. It can be inferred from this phenomenon that, along with increasing concentration, the Eu^3+^ and Tb^3+^ contents and fluorescence intensity also increase because of the increase in the number of luminescence centers. By the time a certain concentration is added, the increase of hydroxyl concentration will lead to fluorescence quenching [32]. By comparing the fluorescence emission intensity of Eu^3+^ only and Tb^3+^ only, the fluorescence emission intensity of Eu^3+^ increases and that of Tb^3+^ decreases. This is because the fluorescence quenching of Tb^3+^ occurs as the Eu^3+^ content increases, and the Eu^3+^ intensity will increase when the Tb^3+^ content increases because of the sensitized fluorescence from Tb^3+^ [33,34].

Figure 7 shows the fluorescence emission intensity of the PVA fluorescent gel for different reaction times by controlling the temperature, reaction concentrations, and Eu^3+^:Tb^3+^ ratios. The figure shows that the intensity of PVA fluorescent gel at the characteristic peak did not change much after 45 min and then decreased slightly after 105 min. Because of the sensitization of Tb, the fluorescence intensity of Eu increased by 117%, 409%, 311%, 237%, 207%, 583%, 138% and 202% (an average of 210%) and the fluorescence intensity of Tb decreased by 58%, 56%, 70%, 54%, 70%, 70% and 77% (an average of 65%). It can be inferred from the above phenomenon that, after the reaction time reaches 45 min, the fluorescence intensity stops rising. The silicon hydroxyl group was hydrolyzed from the ethoxy group in the modified co-doped complexes, so Eu^3+^ and Tb^3+^ are fixed in the PVA gel owing to the fact that the silicon hydroxyl groups bind to the hydroxyl groups in the PVA gel. With the consuming of the complex, the reaction rate decreases, and the increased quantity of the hydroxyl group with time leads to fluorescence quenching due to the excess of complexes.

Figure 8 presents the fluorescence emission intensity of PVA fluorescent gel at different reaction temperatures by controlling the reaction time, reaction concentration, and Eu^3+^:Tb^3+^ ratio. It is shown that the intensity of PVA fluorescent gel at the characteristic peak reaches its maximum when the temperature reaches 80 °C, but the fluorescence intensity decreases gradually with increasing temperature. The collision between complex molecules is more intense, resulting in energy loss, which reduces the energy-transfer efficiency of the ligand to the central ion, reduces the condensation reaction with the hydroxyl group on PVA, and results in the decrease of the fluorescence intensity of PVA fluorescent gel. By calculating the average of fluorescence intensity, the fluorescence intensity of doped europium was 2.4 times higher than before. Terbium, however, exhibited a 45% reduction in fluorescence intensity.

Figure 9 presents the fluorescence emission intensity of the PVA fluorescent gel with different Eu^3+^:Tb^3+^ ratios by controlling the reaction time and reaction concentration. It is shown that the fluorescence intensity of Eu^3+^ first decreased and then increased with the addition of Tb^3+^ when the water content remained unchanged. It can be inferred that Tb^3+^ binds to ligands more easily when the Tb3+:Eu3+ ratio is less than 4 (Eu^3+^ needs four ligands and Tb^3+^ only five); when the Tb^3+^:Eu^3+^ ratio is greater than 4, the sensitization of water to salt is stronger, which leads to an increase in fluorescence.

Figure 10 presents the thermogravimetric (TG) and differential (DTG) curves of PVA gel and PVA fluorescent gel. The thermal decomposition of PVA gel is divided into three stages: (1) removal of water molecules from the PVA; (2) elimination of the branched chain of PVA from the backbone, along with generation of water molecules and acetic acid molecules, with a mass reduction of approximately 51%; and (3) the decomposition of the PVA backbone, producing small molecules, such as acetaldehyde and butenol, with a mass reduction of approximately 18%. After thermal decomposition, approximately 13% of decomposition products remain.

The thermal decomposition process of PVA fluorescent gel is similar to that of PVA gel. However, in the second stage, in addition to water and acetic acid molecules, the Eu/Tb co-doped complex is produced with a mass reduction of approximately 15%, possibly because the complex reduces the amount of branched-chain decomposition by binding to the PVA branched chain. The mass loss in the third stage was approximately 22.5%. After thermal decomposition, approximately 24% of the decomposition products remained, and the excess could be co-doped complexes.

PVA gel began to undergo thermal decomposition at approximately 200 °C, slightly higher than the decomposition temperature of PVA fluorescent gel, indicating that the combination of modified complexes and PVA reduced the thermal stability of PVA gel.

## 4. Conclusions

In summary, a fluorescent gel was prepared by combining rare-earth ions with ligands and using PVA as the carrier. Through the combination of rare-earth elements and ligands, the energy of rare-earth ions is transferred to the central rare-earth ions through the triplet states of the ligands, and the rare-earth ions are stimulated to emit light. The interaction between europium and terbium was studied by doping europium and terbium, and it was found that in the process of energy transfer terbium ions could effectively increase the luminescence intensity of europium ions and play a sensitization role. Meanwhile, an excessive amount of hydroxyl group was found to quench fluorescence. We believe that this series of PVA fluorescent gels has good luminescence and thermal properties and has a certain application value in the field of new energy resources.

## Figures and Tables

**Figure 1 polymers-12-00893-f001:**
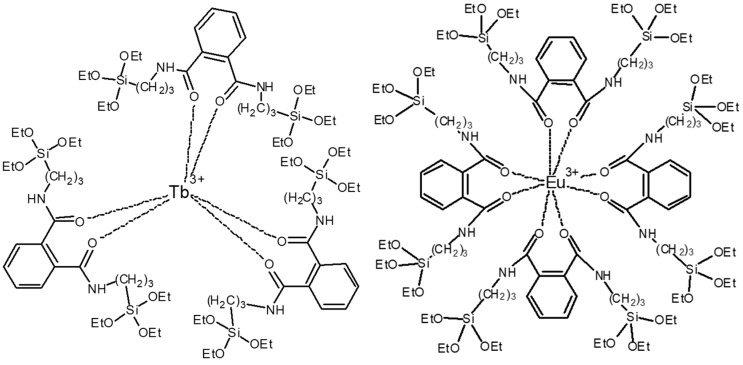
The structural formulas of Tb and Eu complexes.

**Figure 2 polymers-12-00893-f002:**
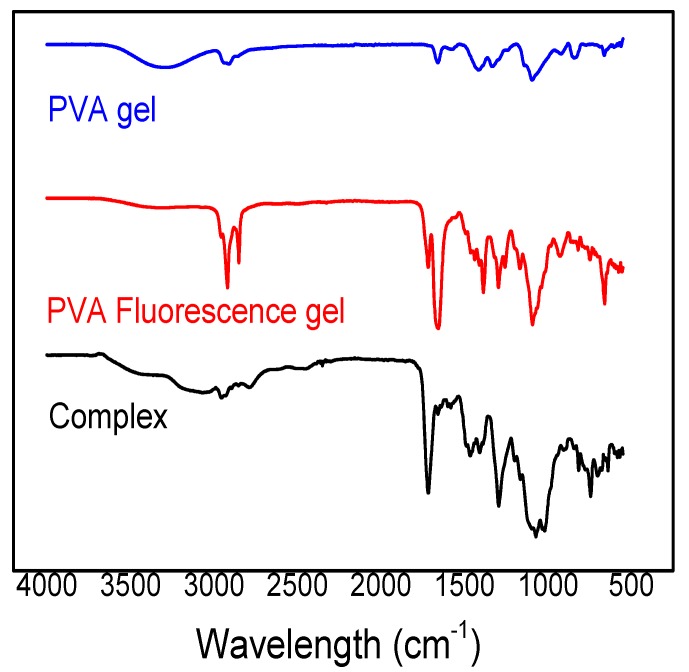
Fourier-transform infrared spectra (FTIR) of modified Eu^3+^/Tb^3+^ complex.

**Figure 3 polymers-12-00893-f003:**
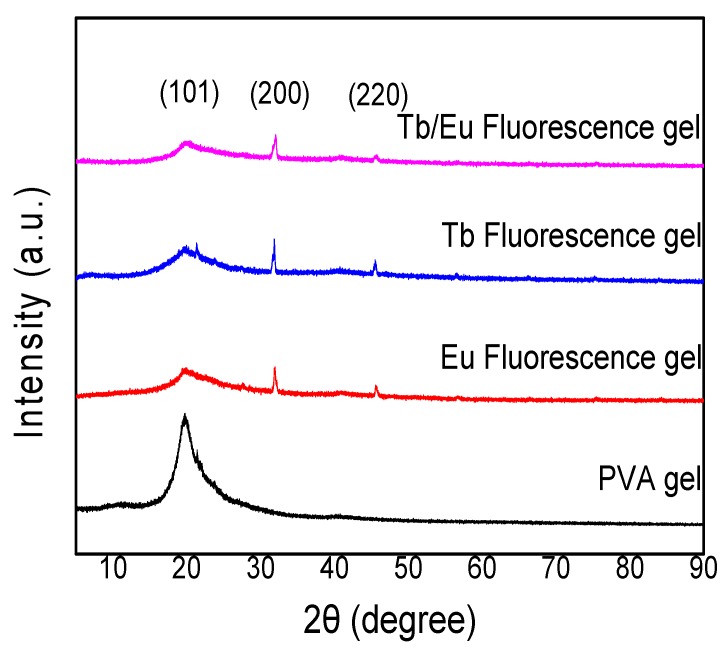
X-ray-diffraction (XRD) curves for PVA gel and Eu, Tb, and Tb/Eu fluorescence gels.

**Figure 4 polymers-12-00893-f004:**
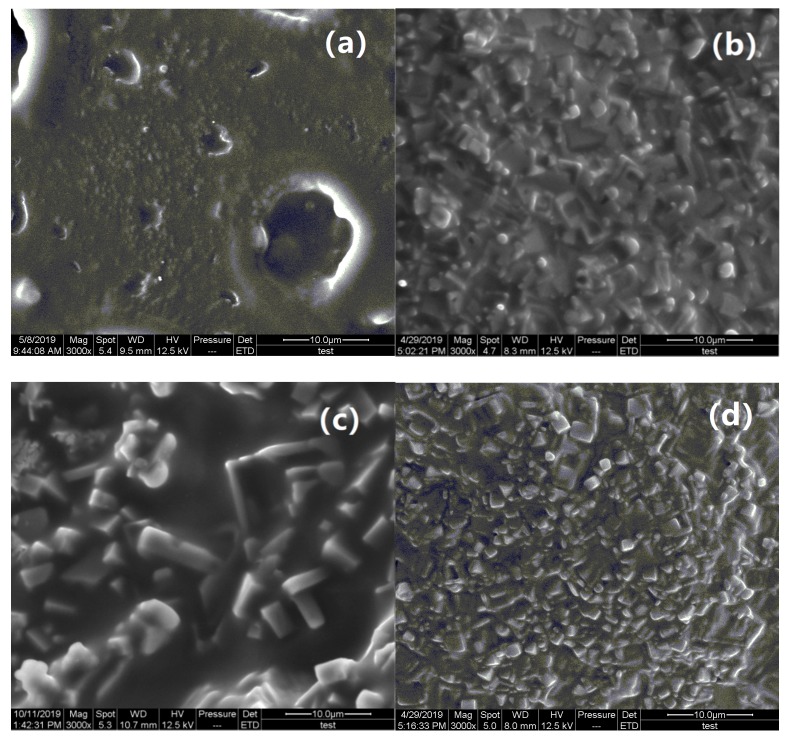
SEM images of (**a**) PVA gel, (**b**) Tb fluorescent gel, (**c**) Eu fluorescent gel, and (**d**) Eu/Tb fluorescent gel.

**Figure 5 polymers-12-00893-f005:**
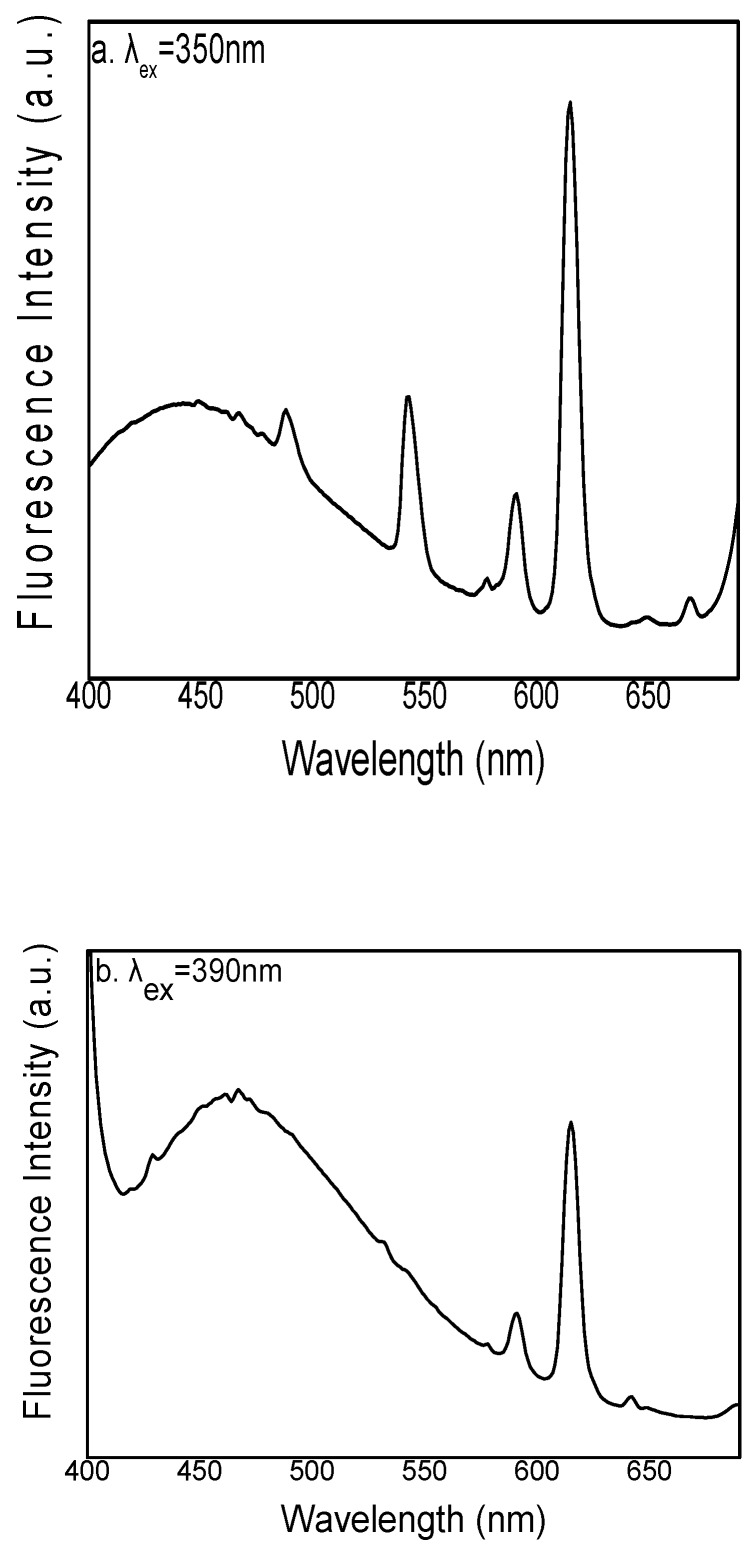
Photoluminescence (PL) excitation spectra of modified rare-earth complexes with wavelengths (**a**) 350 nm, (**b**) 390 nm.

**Figure 6 polymers-12-00893-f006:**
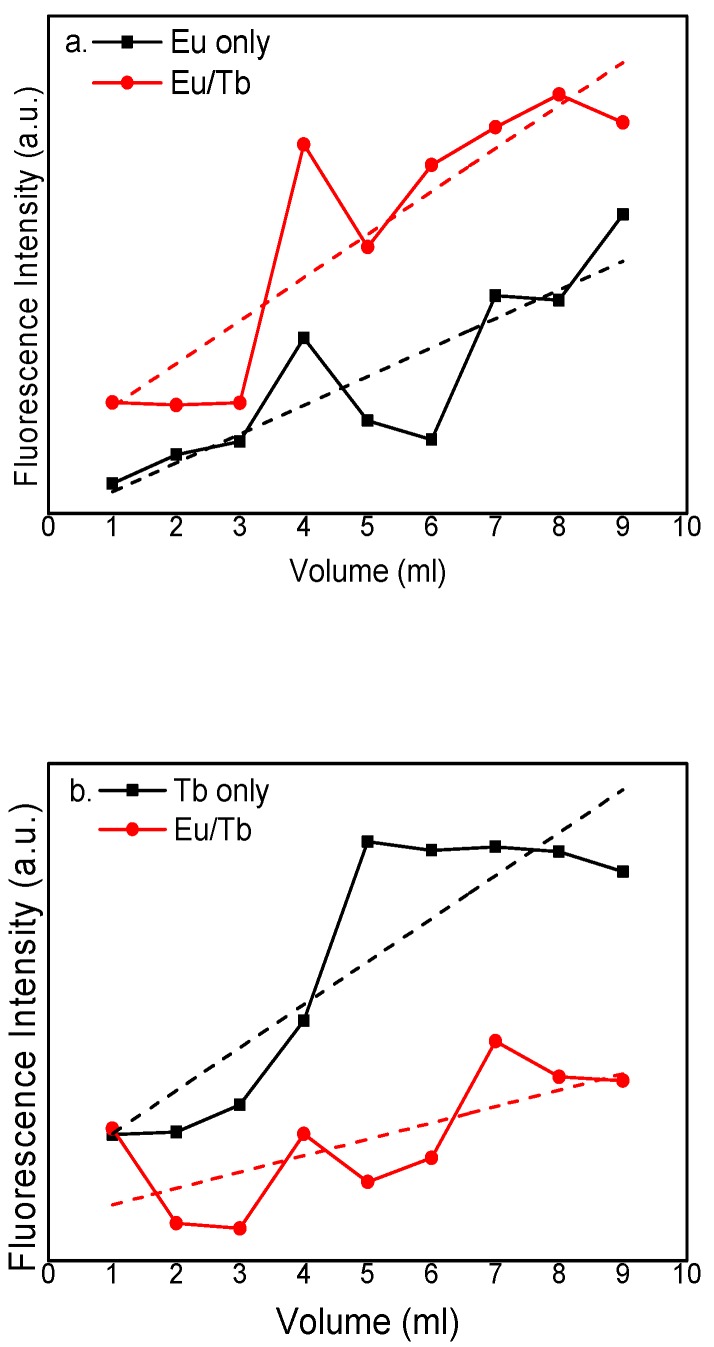
Fluorescence emission intensity of PVA fluorescent gel of (**a**) Eu and (**b**) Tb at different concentrations.

**Figure 7 polymers-12-00893-f007:**
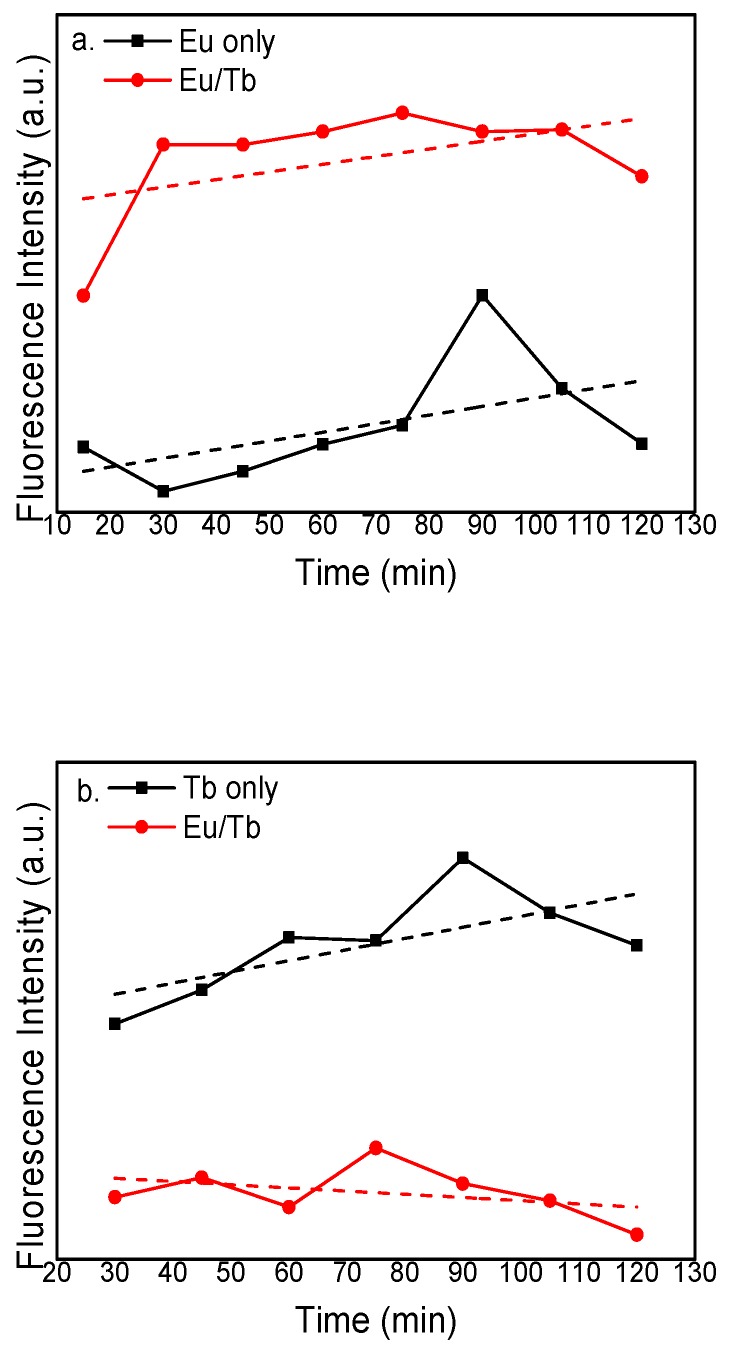
Fluorescence emission intensity of PVA fluorescent gel of (**a**) Eu and (**b**) Tb for different reaction times.

**Figure 8 polymers-12-00893-f008:**
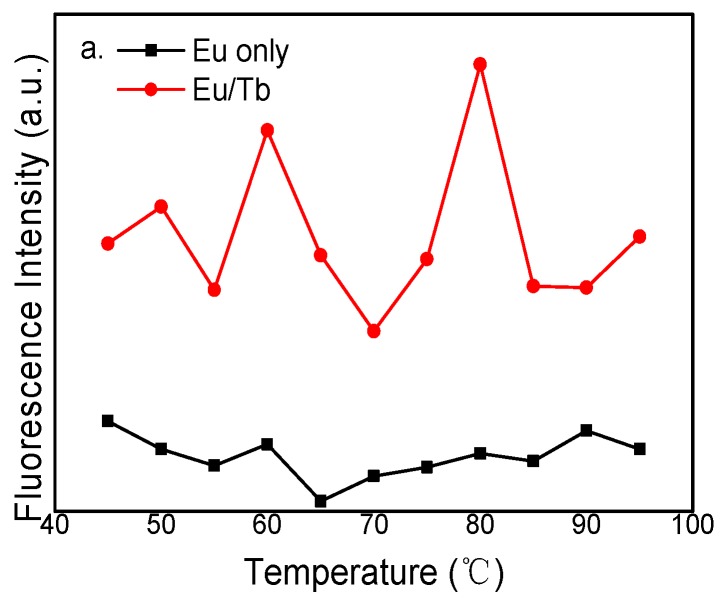
Fluorescence emission intensity of PVA fluorescent gel of (**a**) Eu and (**b**) Tb at different reaction temperatures.

**Figure 9 polymers-12-00893-f009:**
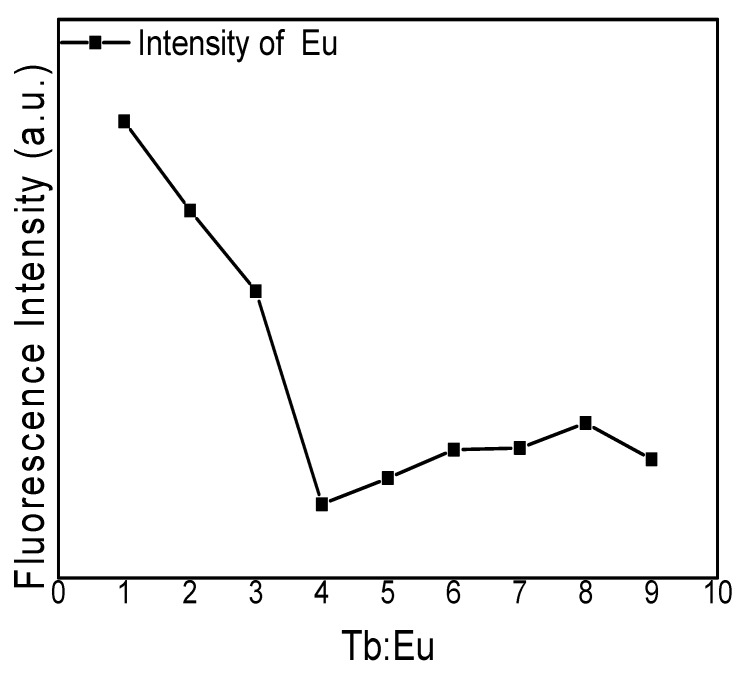
Fluorescence emission intensity of PVA fluorescence gel for different Eu^3+^:Tb^3+^ ratios.

**Figure 10 polymers-12-00893-f010:**
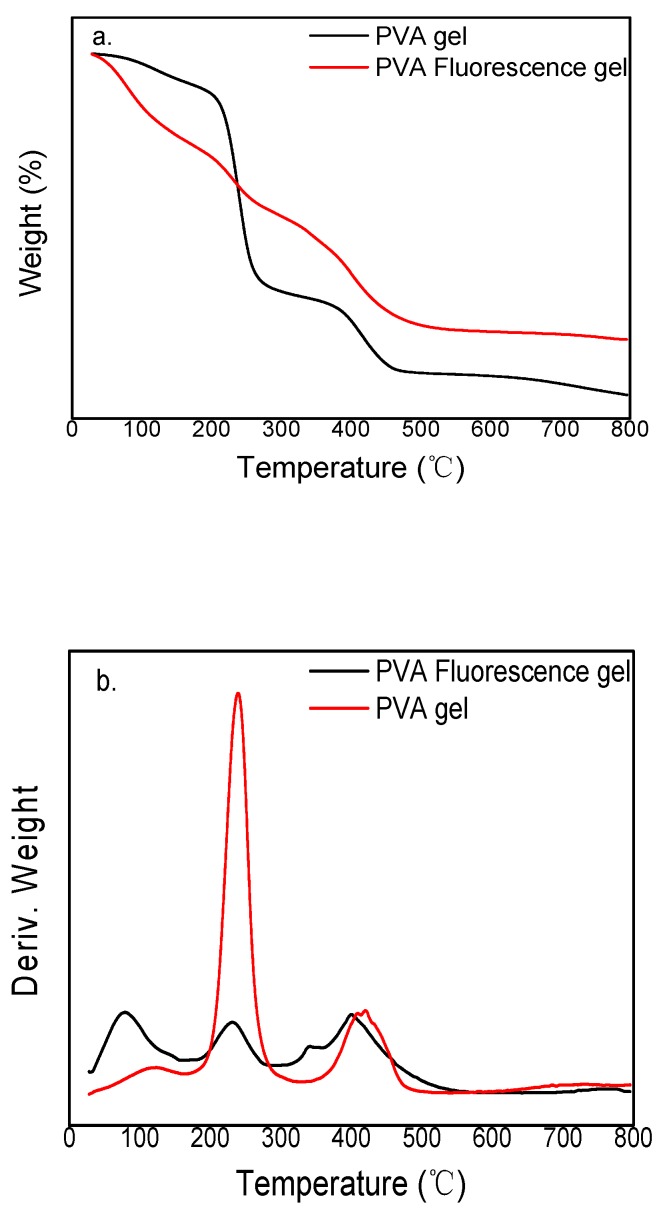
Thermogravimetric (TG) (**a**) and differential thermogravimentric (DTG) (**b**) curves of PVA gel and PVA fluorescent gel.

**Table 1 polymers-12-00893-t001:** Energy-dispersive X-ray-spectroscopy results for Tb/Eu fluorescent gel.

Element	wt.%	at. %
C	51.50	66.85
N	10.96	12.20
O	12.73	12.40
Na	03.68	02.50
Si	03.33	01.85
Cl	07.15	03.14
Eu	03.90	00.40
Tb	06.75	00.66

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
