# Peer review of "Preparation of PVA Fluorescent Gel and Luminescence of Europium Sensitized by Terbium (III)"

_polymers, 2020, doi:10.3390/polym12040893_

Round 1

Reviewer 1 Report

This paper shows the fluorescence characteristic of PVA-BA gel with fluorescence dyes that lanthanide complexes Eu and Tb. Although the data volume is large, the discussion is generally unclear. It is thought that explanations and detailed discussion that are easy for the reader to understand are necessary. Detailed comments are shown as following.

In Introduction

  1. An intro citation of the PVA + boric acid gel used in this experiment is required.
  2. The place of the references 17 looks strange.
  3. The reason for using Eu and Tb in the introduction and for the change the ratio should be shown.
  4. The paper that added Eu and Tb in the PVP gel is cited, but the work should be described in the intro, and the difference and novelty for this work must be show.
  5. As a whole, only rough reviews are cited in introduction, and it is necessary to specifically report studies similar to this study and clarify the novelty.

In experimental

Complex synthesis

  1. KH550 is difficult to understand, so it is necessary to specify the silane coupling agent.
  2. No mention of synthesizing the complex in the intro. References to this complex are also essential. It is also necessary to specify characteristics that have already been investigated.
  3. There may be structural formulas.

Gel synthesis

9.It is necessary to specify that boron is added for gelation and the gelation reaction is performed. Also cited references. What is the necessity to discuss between the reactions?

  1. Specify the container to be enclosed after preparation. It is thought that they are enclosed in different containers according to each analysis. Since there is no detail, the experimental system is unknown. In some cases, studies have been conducted the film by evaporation.
  2. It is necessary to describe the final concentration of the fluorescent substance. If it is mL, I do not know.

Results and Discussion

Line 118

It is unknown where 3.2 comes from. In addition, an error bar is required for the experimental value. If it is not uniform, the error bar will be large.

Line 155

Although it is described that the fluorescence intensity is not linear due to the influence of quenching, it is considered that no sudden change occurs in the case of quenching. The peak in the case of 4 mL cannot be explained. An increase in the fluorescence intensity is also misunderstood by scattering (transparency of the gel). Information on gel transparency and absorption spectra is also needed. Requires error bars for experimental values.

Need more detailed explanation for increase of Luminescence center. It may be in the intro.

Line 162

What is the reaction time in this case? What does mean “A Sufficient Response”

Is the complex broken?

The data in FIG. 6 have many undiscussed results. Also, it is difficult to understand what complex means.

Other

Need information on pH.

Reference

The reference 16 is incorrect.

The reference 28 is incorrect.

From the above, the aim of this research is vague, and there are many parts where the consideration of the results is not enough. Readers have difficulty understanding.

Author Response

Thank you very much for your comments on my article, we have improved it.

Reviewer 2 Report

Thank you for submitting this well written study.

My only criticisms:

  1. the purpose of fluorescent PVA is only mentioned once -- please share some potential applications -- why this is important, as early as possible
  2. pH is known to play a role in molecular fluorescence, while you are studying inorganic fluorescence, you discuss the impact of hydroxyl groups -- it would have been nice if you had listed the pH values of your various formulations

Neither of the above is of great significance, although the first may generate greater interest in the work itself.

Author Response

(The authors gave the same response as above.)

Round 2

Reviewer 1 Report

I was able to confirm that this paper was corrected.